# Hyperbranched Polylysine Exhibits a Collaborative Enhancement of the Antibiotic Capacity to Kill Gram-Negative Pathogens

**DOI:** 10.3390/antibiotics13030217

**Published:** 2024-02-26

**Authors:** Yuxin Gong, Qing Peng, Yu Qiao, Dandan Tian, Yuwei Zhang, Xiaoyan Xiong, Mengxin He, Xiaoqing Xu, Bo Shi

**Affiliations:** 1Feed Research Institute, Chinese Academy of Agricultural Sciences, No. 12 South Zhongguancun Street, Beijing 100081, China; 82101211095@caas.cn (Y.G.); pengqing@caas.cn (Q.P.); qiaoyu@caas.cn (Y.Q.); 82101201074@caas.cn (D.T.); 82101215432@caas.cn (X.X.); 82101212159@caas.cn (M.H.); 2Institute of Agro-Products Preservation and Processing Technology, Tianjin Academy of Agricultural Sciences, Tianjin 300380, China; zhangyuwei2527@163.com

**Keywords:** hyperbranched polylysine, adjuvants, permeabilization, synergistic antibacterial THERAPY

## Abstract

In recent years, traditional antibiotic efficacy outcomes have rapidly diminished due to the advent of drug resistance, and the dose limitation value has increased due to the severe side effect of globalized healthcare. Therefore, novel strategies are required to resensitize resistant pathogens to antibiotics existing in the field and prevent the emergence of drug resistance. In this study, cationic hyperbranched polylysine (HBPL-6) was synthesized using the one-pot polymerization method. HBPL-6 exhibited excellent non-cytotoxicity and bio-solubility properties. The present study also showed that HBPL-6 altered the outer membrane (OM) integrity of *Escherichia coli* O157:H7, *Salmonella typhimurium*, and *Pseudomonas aeruginosa PAO1* by improving their permeability levels. When administered at a safe dosage, HBPL-6 enhanced the accumulation of rifampicin (RIF) and erythromycin (ERY) in bacteria to restore the efficacy of the antibiotics used. Moreover, the combination of HBPL-6 with colistin (COL) reduced the antibiotic dosage, which was helpful in preventing further drug-resistance outcomes. Therefore, this research provides a new strategy for reducing the dosage of drugs used to combat Gram-negative (G^−^) bacteria through their synergistic effects.

## 1. Introduction

Approximately 90% of the antibiotics consumed by individuals globally come from the animals they consume each year, leading to severe problems of bacterial resistance and the presence of drug residues [1]. Many countries have banned the use of antibiotic growth promoters and the application of antibiotics that are important to human medicine in animals [2]. However, antibiotics are still used in the treatment of animals infected by pathogenic bacteria in some countries [3], and the usage is projected to increase by 67% by the year 2030 [4]. A previously conducted study reported that the combination of erythromycin and rifampicin could be used to treat *Rhodococcus equi* pneumonia; however, it was observed that low dosages of combination therapy could induce severe colitis in horses associated with major changes in the intestinal microflora [5]. Intestinal infections caused as a result of Gram-negative (G^−^) bacteria are the main reasons for the use of colistin in livestock [6]. Furthermore, it has been reported in the literature that colistin is the most often prescribed antibiotic used to treat diarrhea caused by *E. coli*, at 40% for pigs and 30% for cattle [7]. The use of colistin in animals has been limited by the emergence of resistant bacteria as the colistin dosage has increased [8]. Therefore, the use of outdated antibiotics that have developed a resistance to these bacteria and the safeguarding of the last line of antibiotics’ defense mechanism are urgent issues that the research must address at present.

In response to these problems, several methods have been developed for the potentization of antibiotics. Due to the factors of biocompatibility and the reduced likelihood of exhibiting resistance, antibiotic adjuvants, such as peptides, polymers, and natural extracts, have been widely used in the field [9]. As an amphiphilic cationic steroid, Ceragenin CSA-13 is a useful agent used to reduce the values of erythromycin MIC used to fight infections caused by antibiotic-resistant *E. coli*, *Pseudomonas* spp., and *S. typhimurium* by more than 8 fold, 8 fold, and 16 fold, respectively [10]. The dual therapy of 7,8-dihydroxyflavone (7,8-DHF) as natural flavonoid-potentiated colistin demonstrated a significantly increased survival outcome (87.5%), which was superior to the outcomes of single-compound therapy of 7,8-DHF and colistin, at 12.5% and 25%, respectively, by converting Fe^3+^ to Fe^2+^ and disrupting the iron homeostasis of *S. typhimurium* [11]. Low concentrations of two synthetic peptides (KLWKKWKKWLK-NH_2_ and GKWKKILGKLIR-NH_2_) used in combination with azithromycin and rifampicin inhibited the growth of most clinical *E. coli*, *K. pneumoniae*, and *A. baumannii* strains [12]. However, these adjuvants have prohibitive costs for large-scale production, as well as metabolic stability concerns and a low application value; the discovery of new natural products is a long-term process [13]. Therefore, it is essential to discover new antibiotic adjuvants.

Recently, different structures of polylysine have been used in the research of antibiotic adjuvants. Poly-L-lysine mainly consists of ε-Poly-L-lysine (ε-PL) and α-Poly-L-lysine (PLL), which are linear chain polymers. Dendritic poly-L-lysine (DPL) is a synthetic nanomaterial, and it has perfect monodispersed macromolecules with a regular, defect-free branched architecture [14]. Hyperbranched polylysine (HBPL) is structurally related to dendritic polylysine, but with a random, branched structure, and both ε-PL and PLL units [15]. ε-poly-L-lysine (n = 25–30) exhibits antimicrobial properties and is commonly used as a food preservative. However, synergistic antimicrobial studies conducted on antibiotics have only shown its potential concerning Gram-positive (G^+^) bacteria. For instance, a previous study showed that metronidazole–ε-PL exhibited a partial synergistic effect on methicillin-resistant *Staphylococcus aureus* (MRSA) and gentamicin–ε-PL exhibited the most powerful antimicrobial effect on methicillin-susceptible *Staphylococcus aureus* (MSSA) [16]. Furthermore, the research presents the bactericidal effects on either sensitive or resistant *P. aeruginosa* in terms of the combination of 2 µM of poly-L-Lysine (pLK) and 1 and 4 µg/mL of imipenem synergies, with a reduction in the bacterial growth properties of both samples displaying up to a 7-log_10_ change compared to the control [17]. However, the stepwise synthesis method is expensive and the product produced has limited solubility and high toxicity properties. HBPL is a highly cationic polymer prepared via the one-step polymerization method with a high yield, which can increase the OM permeability results [18]. Thus, it can be used as an antibiotic adjuvant for the treatment of pathogenic bacteria. The research concerning the application of HBPL to veterinary drugs (for example, rifampicin, erythromycin, and colistin) is still relatively limited.

In this study, polylysine (HBPL-6) synthesized using the one-pot method was successfully obtained. Following a safety evaluation, the structure of HBPL-6 was analyzed by NMR, FTIR, and gel chromatography methods. Furthermore, the permeabilization of pathogens and the combination effect of its use with other antibiotics on the G^−^ pathogens of HBPL-6 were explored in this study. Thus, this paper offers a novel idea for the successful development of medicines, showing the significance of practical research in clinical applications, livestock breeding, and aquaculture industries.

## 2. Results

### 2.1. Characterization of Synthetic HBPL-6

The molecular weight of HBPL-6 was analyzed by GPC, and the result is shown in Figure 1a. The Mn and Mw values of HBPL-6 are 3359 and 13,159 g/mol, and the polydispersity (D = Mw/Mn) value is 3.918, which is similar to the range of chain polymerization products. The ^1^H NMR signal (Figure 1b) of HBPL-6 appears at 4.11 (H_1_), 3.88 (H_2_), 3.36 (H_3_), and 3.26 (H_4_) ppm corresponding to C_α_H protons in dendritic (D), N^α^-linked linear (L_α_), terminal (T), and N^ε^-linked linear (L_ε_) units. After integrating these units, the structural parameter degree of branching (DB) and the average number of branches (ANBs) were calculated as 0.507 and 0.4075, respectively, suggesting that HBPL-6 was a hyperbranched polylysine. The 2D NMR spectroscopy is in Appendix A. The FI-IR absorption spectra of HBPL-6 were detected at 4000–500 cm^−1^, as shown in Figure 1c. For L-lysine hydrochloride, the stretching vibrations of N-H bonds at 3136 cm^−1^ and 2981 cm^−1^ overlap with a strong band in the region of stretching vibrations of C-H bonds. The peak at 1627 cm^−1^ is C=O group. The asymmetric or symmetric deformation vibrations of NH_3_^+^ groups are at 1587 cm^−1^ and 1506 cm^−1^, respectively. The peak at 1405 cm^−1^ may be assigned to the symmetric stretching vibration of the carboxylate group. For HBPL-6, the stretching vibration peak at 3427 cm^−1^ was due to the N-H stretching vibration. The peaks at 2972 cm^−1^ and 2929 cm^−1^ were the stretching vibration peak of -CH_2_. The values of 1637, 1448, and 1375 cm^−1^ corresponded to amide bands Ⅰ, Ⅱ, and Ⅲ, which reflected the characteristic peaks of the C=O stretching, NH_2_ deformation vibration, and C-N stretching vibrations (Figure 1c).

### 2.2. Cytotoxicity and Hemolytic Toxicity of HBPL-6

The cytotoxicity tests we conducted showed that HBPL-6 presented no cytotoxicity values lower than 12.5 μg/mL, and the survival rate reached 100% (Figure 2a). The IC_50_ (the concentration at which the cell viability was reduced to 50%) value was calculated as 24.24 μg/mL, which presented a 50% reduction in the viable counts of Vero cells compared to the buffer treatment. The hemolytic toxicity test presented similar results; 10% of red blood cells were lysed with 50 μg/mL of HBPL-6 (HC_10_ = 50 μg/mL), suggesting an outcome of partial hemolysis, whereas at a value lower than 25 μg/mL, it was less than 5%, indicating an absence of hemolytic toxicity (Figure 2b). Based on these results, it can be observed that the cytotoxicity and hemolytic properties of HBPL-6 generally increase with the concentration.

### 2.3. Bacterial Permeability

In our study, the hyperbranched polylysine synthesized samples for 3 (HBPL-3), 6 (HBPL-6), and 17 h (HBPL-17) were obtained in our preliminary study. The initial evaluation suggested that the NPN uptake factors of HBPL-6 for *S*. *typhimurium* and *E*. *coli* O157:H7 (9.77 and 10.12, respectively) were obviously higher than HBPL-3 (3.4 and 3.28) and HBPL-17 (5.62 and 3.94) at a dose of 0.02 mg/mL, with only a slight difference being evident for *P*. *aeruginosa* PAO1 (-0.72, 0.88, and 0.55) (Appendix A). Hence, HBPL-6 was selected as the object of the present study. When the concentration of HBPL-6 ranged from 0.78 to 100 µg/mL, the NPN uptake factor was increased from 1.25 to 11.86 for *E. coli* O157:H7, 1.61 to 7.16 for *S. typhimurium*, and 1.35 to 10.62 for *P. aeruginosa* PAO1 (Table 1, Table 2 and Table 3), which demonstrated an outcome of dose dependency. Compared to ε-PL at non-inhibitory concentrations ranging from 0.78 to 3.125 µg/mL, it can be observed that the concentration of ε-PL below 6.25 µg/mL presents almost no permeability effect on G^−^ bacteria, while the results demonstrate that the HBPL-6 dose is significantly higher than that of ε-PL at a concentration of 3.125 µg/mL for *E. coli* O157:H7, *S. typhimurium* (*p* < 0.0001), and *P. aeruginosa* PAO1 (*p* < 0.01) (Figure 3a–c, respectively). Therefore, the synthetic HBPL-6 sample was determined to be a more suitable permeability agent than ε-PL.

The PI fluorescence intensity under the administration of 12.5 µg/mL of HBPL-6 was higher than that of the control treated with *S. typhimurium* and *P. aeruginosa* PAO1, and at 25 µg/mL of HBPL-6, it was higher than that of the control treated with *E. coli* O157:H7 (Figure 4a–c). When the addition of HBPL-6 was greater than 12.5 µg/mL (1495.5 ± 19), the PI fluorescence intensity of *S. typhimurium* was highly significantly different compared to the control (688 ± 46.5) (*p* < 0.001) and significantly different (*p* < 0.05) to the concentration of 0.78 µg/mL (897.5 ± 126) (Figure 4b), which indicated that the OM was more significantly damaged when the permeabilizer was added to the incubation stage. The results we achieved show that the addition of 12.5 µg/mL of HBPL-6 caused the nucleic acid to leak in the cytoplasm of *E. coli* O157:H7 and *S. typhimurium* compared to the control (*p* < 0.05) (Figure 4d,e, respectively). For the *P. aeruginosa* PAO1 sample, the OD_260_ was significantly different compared to the control (0.1805 ± 0.009) following the addition of 12.5 µg/mL of HBPL-6 (0.3865 ± 0.016) (*p* < 0.001) (Figure 4f). In addition, HBPL-6 triggered the accumulation of ROS in all the bacteria, which correspondingly aggravated the membrane damage that occurred to further synergize the sterilization (Figure 4g,h,i).

### 2.4. MIC Determination and Checkerboard Synergy Test

#### 2.4.1. MIC Values of HBPL-6

The MICs of the synthesized HBPL-6 used for *S*. *typhimurium*, *E*. *coli* O157:H7, and *P*. *aeruginosa* PAO1 were tested separately, and it was observed that there was approximately a 20% inhibition rate at 100 μg/mL. However, at 2500 μg/mL, it did not inhibit 100% of the G^−^ bacteria, and at a 25 μg/mL concentration, it completely inhibited *Staphylococcus aureus* (Table 4). As the results show, it is not a bactericidal agent and is therefore less likely to be resistant to G^−^ organisms when assisting antibiotics.

#### 2.4.2. Synergistic Effect Analysis of HBPL-6 on Antibiotics

The effects of antibiotics used alone and the combination of HBPL-6 with rifampicin, erythromycin, colistin, minocycline, tetracycline, gentamicin, neomycin, tobramycin, amikacin, and ampicillin on representative G^−^ bacterial infections (*S. typhimurium*) were evaluated through the MIC test (Table 5). As a result, the HBPL-6 and half of the antibiotics’ combined effects were greater than the sum of their respective separate activities. The FICI values were 0.03125 and 0.125 for the combination of HBPL-6-erythromycin and HBPL-6-rifampicin; thus, these results reveal the synergistic effect (FICI ≤ 0.5) of the combination. The FICI value was 0.5 for the combination of HBPL-6-colistin, HBPL-6-minocycline, and HBPL-6-tetracyclin. The results also demonstrate that the activities of these antibiotics are enhanced by the use of HBPL-6. However, the HBPL-6 combination of aminoglycoside antibiotics, such as gentamicin, neomycin, tobramycin, and amikacin, presented an indifferent effect on *S. typhimurium* with an FICI value equal to 1. Consequently, HBPL-6 has the potential to be used as a cationic permeabilizing agent for some hydrophobic antibiotics, such as erythromycin and rifampicin, which are blocked by the cell membrane. Moreover, the combined effect reduced the MIC values of colistin, minocycline, and tetracycline to 1/2. Moreover, other samples of G^−^ bacteria are being studied in relation to older drugs with significant synergistic effects (rifampicin and erythromycin), and they are the last line of defense for antibiotics (colistin) that are being reintroduced into the market at present.

The effect of the co-administration of HBPL-6 and erythromycin treated on *E*. *coli* O157:H7, *S*. *typhimurium*, and *P*. *aeruginosa* PAO1 was mapped on the experimental response surface, which presented synergy effects (Figure 5a–c, respectively). The combination growth curve values of 6.25 and 12.5 μg/mL for HBPL-6 used in combination with 8 μg/mL of erythromycin revealed that the use of 12.5 μg/mL of HBPL-6 resulted in the complete inhibition of *E*. *coli* O157:H7 compared with control, monotherapy, and dual-therapy scenarios (Figure 5d,g). When 6.25 and 25 μg/mL concentrations of HBPL-6 were used in combination with 16 μg/mL of erythromycin, there was no effect evident on *S*. *typhimurium* at 6.25 μg/mL and no decrease in the MIC, and the use of 25 μg/mL of HBPL-6 resulted in the complete inhibition of *S*. *typhimurium* at that antibiotic concentration (Figure 5e,h). When 12.5 and 25 μg/mL doses of HBPL-6 were administered in combination with 16 μg/mL of erythromycin, significant retardation of *P*. *aeruginosa* PAO1 growth at 12.5 μg/mL until 15 h was evident, and 25 μg/mL of HBPL-6 worked with the antibiotic to result in the complete inhibition of *P*. *aeruginosa* PAO1; a change in the MIC was also evident (Figure 5f,i). 

The effect of the co-administration of HBPL-6 and rifampicin to treat *E*. *coli* O157:H7, *S*. *typhimurium*, and *P*. *aeruginosa* PAO1 were mapped on the experimental response surface, which presented the synergy effects (Figure 6a–c, respectively). The combination growth curve values of 6.25 or 12.5 μg/mL for HBPL-6 used in combination with 8 μg/mL of rifampicin revealed that a 12.5μg/mL dose resulted in the complete inhibition of *E*. *coli* O157:H7 compared with the control, individual drugs, and compound drugs (Figure 6d,g). When 6.25 doses and 12.5 μg/mL of HBPL-6 were administered in combination with 2 μg/mL of rifampicin, the growth of *S*. *typhimurium* was significantly delayed without causing the MIC to decrease at a 6.25 μg/mL HBPL-6 dose. At a 12.5 μg/mL dose, the antibiotic completely inhibited *S*. *typhimurium* (Figure 6e,h). The 6.25 and 25 μg/mL doses of HBPL-6 administered in combination with 16 μg/mL of rifampicin showed no obvious effect at 6.25 μg/mL, and HBPL-6 at 25 μg/mL delayed the growth rate of *P*. *aeruginosa* PAO1 (Figure 6f,i).

The effects of the co-administration of HBPL-6 and colistin used to treat *E*. *coli* O157:H7, *S*. *typhimurium*, and *P*. *aeruginosa* PAO1 were mapped on the experimental response surface, which presented the synergy effects (Figure 7a–c, respectively). The combination growth curve values of either 3.125 or 6.25 μg/mL of HBPL-6 used in combination with 0.25 μg/mL of colistin revealed that 3.125 μg/mL of colistin retarded the growth of *E. coli* from 8 h, and a dose of 6.25 μg/mL was fully synergistic (Figure 7d,g). When 0.78 and 1.56 μg/mL doses of HBPL-6 were used in combination with 1 μg/mL of colistin, a 0.78 μg/mL dose allowed colistin to retard the growth of *S*. *typhimurium* 12 h later; a 1.56 μg/mL dose was fully synergistic in the inhibition phase, converting to a 1/2 MIC (Figure 7e,h). The combination of a 6.25 μg/mL colistin dose resulted in the retardation of *P*. *aeruginosa* PAO1 growth from 9 h, and a 12.5 μg/mL dose resulted in a complete synergistic inhibition result of 1/2 MIC (Figure 7f,i). However, in the positive control group, the two combinations presented antagonistic behaviors when treating *S*. *typhimurium* with ε-PL and colistin (Appendix A).

SEM analysis was used to present the changes occurring on the surface of *S. typhimurium* during different treatments. The bacteria in the control group had rod-shaped cells with relatively smooth surfaces and intact cell membranes (Figure 8a). No bacterial changes were observed in the FICI values of colistin (Figure 8c). However, the surface wrinkling occurred following the addition of 12.5 μg/mL of HBPL-6 (Figure 8b). For the combined treatment, the alterations occurred on the bacterial surface and the cell membrane ruptured with pores (Figure 8d). The results indicate that the permeabilizer serves to disrupt the bacterial outer membrane and further synergizes with the bactericidal effect of colistin.

## 3. Discussion

During the synthesis process, the reaction time and temperature were important factors we considered, which affected the properties of certain thermal polymerization factors, such as gelation, structure, and yield. As previously reported, the mixture under study was stirred for 48 h at 160 °C, and the water was removed with nitrogen for 2 min at a pressure of 0.15 bar after a 24 h period (Mn = 9400, PD = 1.48, DB = 0.5) [19]. In contrast to the HBPL-6 sample, the difference we observed was that it was below a −0.10 MPa reaction value for 6 h (Mn = 3359, PD = 3.918, DB = 0.507), and low-molecular-weight polylysine (Mn < 10 kDa) samples were targeted as a lower molecular weight decreases the cytotoxicity and improves the solubility in water. Additionally, in another study, the sample was heated up to 240 °C for 5 h in a ceramic crucible, and the DB value achieved was 0.4 [20]. In this study, it was observed that the open microwave reaction, in comparison to the thermal polymerization method, was expectedly faster—the Mn value was 4200 at 0.5 h at 200 °C—however, this approach was more costly to use. HBPL-6 presented a greater polydispersity value and a random branch and was more water soluble with a shorter reaction time and lower temperature. The ^1^H NMR signals of the HBPL-6 sample were similar to those previously reported as 4.13, 3.85, 3.33, and 3.23 ppm corresponding to C_α_H protons in D, L_α_, T, and L_ε_ units [21]. The FTIR results show that the structure of lysine monomer is present in which amino groups are protonated owing to the carboxyl group and HCl acid [22]. In the polymer HBPL-6, it contained the amide group (-CONH) [23,24]. There are changes between mono-lysine HCl and polymer HBPL-6, so it can thus be further proved that hyperbranched polylysines were synthesized.

In the macromolecular synthesis of a dual-drug interaction, the factors of effective concentration and dose-limiting toxicity are a serious concern. HBPL-6 is a cationic polymer with numerous amine groups. Therefore, the number and arrangement of the cationic charges in HBPL-6 affect and interact with cell membranes, and then present cell-damaging effects [25]. As can be observed by the same results achieved in this study, a high concentration of HBPL-6 leads to both cytotoxicity and hemolytic toxicity effects (IC_50_ = 24.24 μg/mL, HC_10_ = 50 μg/mL) with Mn = 3359. A previous study demonstrated that the EC_50_ values of the HBPL samples ranged from 10 to 1 mM, where the Mn values ranged from 1400 to 146,800, suggesting greater cell-damaging effects with the increasing molecular mass of the polymer [26]. Toxicity leads to dosage limitations in the application of HBPL-6; therefore, it is essential to reduce toxicity levels. Previous studies have shown that hyperbranched random co-polymers reduced cytotoxicity and increased selectivity between bacteria and mammalian cells by copolymerizing lysine with a hydrophobic amino acid, e.g., alanine, tryptophan, or phenylalanine [27]. In conclusion, the molecular mass, charge, and structure are the main factors that determine the biological toxicity levels of HBPL-6. Follow-up experiments should focus on reducing the toxicity levels and the interactions with antibiotics by altering the properties of HBPL-6.

Polylysine is an important class of polyamino acid with three basic structures [28]. Compared with α-polylysine’s high toxicity level, ε-PL is produced by the microbial synthesis process as a class of natural polymers and is widely used in various foods and medicines, including antibiotics [29]. Another major polylysine is the hyperbranched polymer (HBPL-6), which was synthesized in this study. In order to further investigate whether HBPL-6 also has the ability to synergize antibiotics as commonly used additives, a comparison of the permeabilization effects between HBPL-6 and ε-PL was performed. The results indicate that HBPL-6 is more suitable for use as a permeating agent. As the previous studies show, the relative fluorescence values of NPN by *E*. *coli* suspensions treated with gallic acid–g-chitosan (I) at MIC, 2 × MIC, and 4 × MIC are 18, 22, and 32, respectively, similar to HBPL-6’s dose dependence value [30]. Primary amines are evident at the C-2 position on chitosan and at the C-2 and C-6 positions on HBPL-6. Thus, polycationic antimicrobial agents can bind to the negatively charged O-specific oligosaccharide units of lipopolysaccharides (LPSs) and phospholipids, thus disrupting the integrity of OM. Moreover, the antibacterial peptide (CM4) has the ability to neutralize LPSs from *E*. *coli* 0111: B4. CM4 was observed to inhibit the LPS-induced activation of Limulus amoebocyte lysate in a dose-dependent manner, and a reduction in the bactericidal activity of CM4 was also observed as the concentration of the LPS increased. This also confirmed the affinity of the cationic peptide for LPSs [31]. As previously reported, polycationic compounds, such as protamine, induced the leakage of cytoplasmic components from *Listeria monocytogenes* and *E. coli* cells by interacting with the cell surface membrane [32]. At the MIC concentration of ε-PL, the production rate of ROS accounted for a value of approximately 93.7% [33]. Based on the results achieved in this study, it can be concluded that the synergistic mechanism of HBPL-6 is the physical disruption of the cytoplasmic membrane following electrostatic absorption activity.

G^−^ bacteria exhibit high levels of intrinsic resistance to clinically relevant G^+^ antibiotics (e.g., rifampicin and erythromycin), which occurs primarily due to the impermeability of the bacterial outer membrane with LPSs and a polyanionic core (Mg^2+^, Ca^2+^) [34]. Therefore, to restore the functions of such antibiotics, the neutralization of surface anions with cationic adjuvants is required. In a previous study, the fractional bactericidal concentration index (FBCI) of CATH-1 combined with erythromycin was investigated and presented as 0.125. However, the FBCIs of ampicillin, tetracycline, and gentamicin were 1.031, 0.75, and 0.75 when used to combat *E*. *coli* O157:H7 [35]. These results demonstrate that the combined use of erythromycin presents good synergistic antibacterial activity. In this study, the FBCIs of HBPL-6 combined with erythromycin were 0.75, 0.625, and 0.5625 when used to target *E*. *coli* O157:H7, *S*. *typhimurium*, and *P*. *aeruginosa* PAO1, respectively, which presented better synergy outcomes than the recently achieved results. In another study, concerning the synergistic effect of cationic polyurethane combined with rifampicin, a polyketide was synthesized from a molar ratio of a Boc-protected Lys diol monomer and OTBS-protected diol monomer, reducing the MIC of rifampicin in *E. coli* by up to 64 fold [36]. However, the cationic pendant group Lys was synthesized and then subjected to multi-step reactions, such as catalysis and deprotection for polymerization purposes. While the synthesis of HBPL-6 employs a one-pot method, L-(+)-Lysine monohydrochloride and KOH are relatively simple and readily available precursors [37]. The growth of other G^−^ bacteria, such as *Acinetobacter baumannii*, can be inhibited by 96% following a combination of 4.0 μg/mL of SPR741 and 1.0 μg/mL of rifampin, with a minimum four-fold reduction in most MICs [38]. In this study, compared with rifampin and erythromycin, the combination of colistin and HBPL-6 was the most effective method. We observed that, with the lowest test dose (0.78 μg/mL), the best synergistic effect was achieved. Numerous AMPs have been used in the research as antibiotic adjuvants for colistin synergy purposes. While fewer studies exist on the aspect of polycation synergism, this study provides technical support to argue for the reduction in the use of colistin and guard the last line of defense of antibiotics. However, it also shows the indifferent effects of aminoglycoside antibiotics, e.g., gentamicin, neomycin, tobramycin, and amikacin, when combined with BHPL-6 (FICI = 1). Recently conducted research demonstrates that appending hydrophobic moieties onto aminoglycosides, like tobramycin, can generate amphiphilic aminoglycosides with weak or no protein translation inhibitory effects, but can also exhibit potent OM-disrupting and IM-uncoupling properties against *P. aeruginosa* [39,40]. Based on the results of this study, we speculated that the possible cause of the indifferent effect was that the outer-membrane disruption of tobramycin could be uncoupled from its ribosomal effects [41]. Consequently, the efficacy of antibiotics was simultaneously counteracted and augmented, demonstrating no apparent synergistic effect.

## 4. Materials and Methods

### 4.1. Materials

The following pathogenic bacteria were used in the tests: *Salmonella typhimurium* (CICC 22956, China Industrial Microbial Strain Preservation and Management Centre), *Escherichia coli* O157:H7 (CICC 10907, China Industrial Microbial Strain Collection Management Centre, Beijing, China), and *Pseudomonas aeruginosa* PAO1 (ATCC 27853, American Type Culture Collection, Littleton, CO, USA). The following cells were used in our experiments: African green monkey kidney Vero cells (ATCC CCL-81, Wuhan Punosai Life Science and Technology Co., Wuhan, China) and sheep blood cells (fresh, sterile, defibrinated sheep blood, Beijing Landbridge Technology Co., Beijing, China). The following materials were used in our experiments: L-(+)-Lysine monohydrochloride (C_6_H_14_N_2_O_2_∙HCl) (CAS:657-27-2, Macklin) and potassium hydroxide (KOH) (GB 2306-80, Beijing Chemical Factory, Beijing, China). The following antibiotics were used in our experiments: colistin (CAS: 1264-72-8), erythromycin (CAS: 114-07-8), and rifampicin (CAS: 13292-46-1).

### 4.2. Synthesis of Hyperbranched Polylysine-6h (HBPL-6)

L-(+)-Lysine monohydrochloride (33 g, 0.18 mol) and KOH (9.9 g, 0.18 mol) were thoroughly mixed with 10 mL of distilled water [21]. The reaction was performed at 160 °C with agitation (600 rpm). After we achieved a viscous product, the reaction was continued for 6 h at −0.10 MPa. Distilled water (200 mL) was added to the mixture, and the product was dissolved by the sonication method for 30 min after being cooled down to room temperature. The reaction solution was filtered through a Büchner funnel to remove an insoluble brownish-black residue. The filtrate was then dialyzed using a dialysis bag at 500 Da (Biotopped Corporation, Viskase, Lombard, IL, USA) for 72 h in a water system, and any lost water was replaced every 12 h. The dialyzed liquid was concentrated to 10 mL using a rotary evaporator and then lyophilized. The powder achieved was stored for the following analysis.

### 4.3. Characterization Measurement of HBPL-6

The molecular weight was determined by gel chromatography on a TSKgel 2000 SW_XL_ measuring 300 mm × 7.8 mm with acetonitrile/water/trifluoroacetic acid (*v*:*v*:*v* = 40:60:0.1) as the mobile phase equipped with an ultraviolet detector. The detection procedure was performed at 220 nm, the flow rate was 0.5 mL/min, and the column temperature was 30 °C. Cytochrome C, aprotinin, bacitracin, GLY-GLY-TYR-ARG, and GLY-GLY-GLY were used as the standards, and the concentration of the standards was 0.1 mg/mL.

^1^H NMR spectra were recorded at 25 °C on a Bruker Avance-500 NMR spectrometer (Bruker Corporation, Karlsruhe, Germany). Deuterium oxide (D_2_O) was used as the solvent. The samples were dissolved in 0.6 mL of D_2_O and transferred to a 5 mm NMR sample tube. CD_4_O was used as an internal standard. The degree of branching (DB) and the average number of branches (ANBs) were calculated using the integrals of the different structural units in the ^1^H spectra [20]. The DB and ANBs were calculated according to the following formula:$$DB=\frac{D+T}{D+L+T}=\frac{D+T}{D+L\alpha +L\varepsilon +T}$$$$ANB=\frac{D}{D+L}$$

Fourier transform infrared (FTIR) spectra were tested on a Bruker Avance, model VERTEX 70V; HYPERION 2000 (Bruker Corporation, Karlsruhe, Germany). The spectral range was 4000–500 cm^−1^, the resolution was 0.16 cm^−1^, the wave number accuracy was 0.01 cm^−1^, and the light transmission accuracy was better than 0.07%T.

### 4.4. Cytotoxicity of HBPL-6

The cytotoxicity of HBPL-6 was determined using an African monkey kidney cell model (Vero cells) with the 3-(4,5-dimethylthiazol-2-yl)-2,5-diphenyltetrazolium bromide (MTT) method (VWR, Lutterworth, UK) [42]. The cells were incubated in Dulbecco’s Modified Eagle’s Medium (DMEM) supplemented with 10% fetal bovine serum (FBS) and 1% penicillin–streptomycin (P/S) at 37 °C in a humidified environment with 5% CO_2_. The cells were detached from the dishes using 0.25% trypsin–ethylenediaminetetraacetic acid (2.5 g/L of trypsin and 1 g/L of EDTA); the suspensions of the cells were seeded in a 96-well plate (1 × 10^4^ cells/well; 100 μL/well) and cultivated for 24 h. After removing the cell supernatant from the 96-well plate, the 100 μL/well of the HBPL-6 solution at different concentrations (1.56, 3.125, 6.25, 12.5, 25, 50, and 100 μg/mL, dissolved in a cell culture medium) was added and incubated for 24 h. After the incubation stage, the cells were first incubated for 2 h with MTT and then with isopropanol for 30 min. The OD_570_ was detected to calculate the cell viability value. 

### 4.5. Hemolytic Toxicity of HBPL-6 

The hemolytic activity of HBPL-6 was determined using sheep hemocytes prepared from fresh, sterile, defibrinated sheep blood (Landbridge Technology, Beijing, China) [43]. The sheep hemocytes were treated with different concentrations (at the values presented above) for 4 h at 37 °C. Phosphate-buffered saline (PBS: 0.01 mol/L; pH 7.4) and 0.2% Triton X-100 were used as the positive and negative controls, respectively. The absorption value of the released hemoglobin was measured at 576 nm by an enzyme-labeled instrument.

### 4.6. The Minimum Inhibitory Concentration of HBPL-6 on G^−^ Bacteria

The minimum inhibitory concentration (MIC) test utilized the broth dilution method as described by the American Clinical Laboratory Standards Institute (CLSI) [44]. Fresh Mueller–Hinton broth medium (MHB) was inoculated with *S*. *typhimurium* (CICC 22956), *E. coli* O157:H7 (CICC 10907), and *P. aeruginosa* PAO1 (ATCC 27853) and shaken at a speed of 180 rpm at 37 °C for 12 h. The cells were then harvested in 96-well plates to produce an initial cell density of approximately 1 × 10^6^ CFU/well and incubated, with increasing concentrations of HBPL-6 (from 0 to 2500 μg/mL) detected every 30 min for 18 h of incubation in a thermostatic enzyme-labeled instrument at 37 °C. Each assay was performed at least three times.

### 4.7. Bacterial Permeabilization of Polylysine 

*E. coli* O157:H7, *S*. *typhimurium*, and *P. aeruginosa* PAO1 were cultured to OD_630_ = 0.5 ± 0.02, centrifuged at 1000× *g* for 10 min; the cells were then suspended in 5 mmol/L of the HEPES (pH = 7.2) buffer, and the OD_630_ was adjusted to 1.0. The 96-well plates were supplemented with 200 μL of the reaction agents, including 100 μL of bacterial suspension, 50 μL of 40 μmol/L N-phenyl-1-naphthylamine (NPN) (dissolved in HEPES buffer), and HBPL-6 or ε-PL in 50 μL of HEPES buffer to a final concentration that ranged from 0.78 to 100 μg/mL. Control wells were prepared as follows: (i) HEPES buffer alone (200 μL); (ii) HEPES buffer (150 μL) and NPN (50 μL); (iii) bacterial suspension (100 μL) and HEPES buffer (100 μL); and (iv) bacterial suspension (100 μL), NPN (50 μL), and HEPES buffer (50 μL). ε-Poly-L-lysine (ε-PL) was used as the control to analyze the permeability of HBPL-6. The values were recorded within 3 min. Each assay was performed at least three times [45]. 

### 4.8. Membrane Integrity Effect of HBPL-6

The bacterial suspensions of *E. coli* O157:H7, *S*. *typhimurium*, and *P. aeruginosa* PAO1 were washed and resuspended in 1 × PBS (pH 7.4) to obtain an OD_600_ = 0.5, followed by the addition of 10 nmol/L of fluorochrome propidium iodide (PI) (Sigma-Aldrich, Shanghai, China, no. P4170). For the addition of HBPL-6 (0–100 μg/mL), the fluorescence was measured after an incubation period of 30 min. In the presence of HBPL-6, the fluorescence was measured every 10 min with the excitation wavelength at 535 nm and emission wavelength at 615 nm during an incubation period of 90 min.

### 4.9. Nucleic Acid Leakage in the Cytoplasm Treated with HBPL-6

*E. coli* O157:H7, *S*. *typhimurium*, and *P. aeruginosa* PAO1 were initially incubated overnight at 37 °C. Following the centrifugation of the bacterial culture at 4000× *g* for 15 min, the pellets were washed twice with 1 × PBS (pH 7.4). The bacterial suspensions were treated with the addition of HBPL-6 (0–100 μg/mL). The control was a suspension that only contained bacteria and PBS. All the samples were incubated for 3 h at 37 °C and centrifuged at 13,400× *g* for 15 min at the end of the incubation period to collect the supernatant. The OD_260_ was detected to determine the amount of nucleic acid released from the cytoplasm. Each test was performed at least three times [46].

### 4.10. Reactive Oxygen Species (ROS) Measurement in Bacteria

The levels of ROS in *E. coli* O157:H7, *S. typhimurium*, and *P. aeruginosa* PAO1 after being treated by HBPL-6 (0–100 μg/mL) were also determined by fluorescence spectrophotometry. The suspensions were washed and resuspended in 1 × PBS (pH 7.4) to obtain an OD_600_ = 0.5 and then incubated with 10 μmol/L of 2’,7’-dichlorofluorescein diacetate (DCFH-DA) (BiYunTian, Beijing, China) at 37 °C for 30 min. After being washed with PBS 3 times, 190 μL of probe-labeled bacterial cells was added with 10 μL of HBPL-6. After incubation for another 30 min, the fluorescence intensity was measured with the excitation wavelength at 488 nm and the emission wavelength at 525 nm [47].

### 4.11. Synergistic Potentiation of HBPL-6 on Antibiotics

The chequerboard assay was used to assess the combination effect of compounds with antibiotics and to calculate the potentiating activity [48]. First, *E. coli* O157:H7, *S*. *typhimurium*, and *P. aeruginosa* PAO1 were grown on the solid MHB medium, and then single colonies were inoculated in the MHB medium and incubated at 37 °C at 180 rpm/min. The suspension was adjusted to 10^6^ CFU/mL. The 96-well plates were supplemented with 200 μL of the reaction agent, including 50 μL of antibiotics (erythromycin, rifampin, or colistin) dissolved in MHB, 50 μL of the bacterial suspension, 50 μL of HBPL-6 dissolved in MHB, and 50 μL of MHB. The final concentrations of colistin were 16, 8, 4, 2, 1, 0.5, 0.25, and 0.125 μg/mL; the final concentrations of erythromycin were 128, 64, 32, 16, 8, 4, 2, and 1 μg/mL. The final concentrations of rifampicin were 64, 32, 16, 8, 4, 2, 1, and 0.5 μg/mL; the final concentrations of HBPL-6 were 100, 50, 25, 12.5, 6.25, 3.125, 1.56, 0.78, and 0.39 μg/mL; and the final concentrations of other antibiotics (ampicillin, gentamicin, tetracycline, minocycline, neomycin, tobramycin, and amikacin) ranged from 0 to 1024 μg/mL. The OD_600_ value of each well was detected every 30 min for 18 h of incubation at 37 °C in a thermostatic microplate reader (BioTek Synergy H1, Winooski, VT, USA). The growth curve was then plotted. The response surface plots for the drug conjugations of erythromycin, rifampin, and colistin were drawn using Combenefit software version 2.02. Each assay was performed at least three times. The fractional inhibitory concentration (FICI) of the antibiotic was calculated according to the following formula:$$FICI=\frac{{MIC}_{AB}}{{MIC}_{A}}+\frac{{MIC}_{BA}}{{MIC}_{B}}$$

### 4.12. Scanning Electron Microscopy

The SEM examination was carried out to observe the potential impacts of HBPL-6 or colistin alone, as well as the synergistic combinations (HBPL-6 + COL) on the cell morphology of *S. typhimurium*. Bacteria were incubated at 37 °C overnight in MHB medium and then treated with HBPL-6 (12.5 µg/mL), colistin (1 µg/mL), synergistic combinations of HBPL-6 and colistin, and without any additions as the control. The samples were centrifuged for 10 min at 4000× *g* and then fixed in 2.5% glutaraldehyde at 4 °C for 12 h after being suspended 3 times with deionized water for 6, 7, and 8 min in succession. The bacterial pellets were then dehydrated using a graded ethanol series (50, 70, 85, 95, and 100%) for 15 min. The samples were dried and coated with gold using the sputtering method under a vacuum and examined under a scanning electron microscope (SU8010, Hitachi, Japan).

### 4.13. Statistical Analyses

Three independent replications were conducted for each treatment. Statistical analyses were performed using GraphPad Prism 8.0 (SAS Institute, Inc., Cary, NC, USA). ANOVA and *t*-test multiple comparisons were used and the data were assumed to be statistically significant at *p* < 0.05. (n.s.: not significant; *: *p* < 0.05, **: *p* < 0.01, ***: *p* < 0.001, ****: *p* < 0.0001).

## 5. Conclusions

HBPL-6 was synthesized as a hyperbranched polycation using the one-pot method, which presented low toxicity, high solubility, and high yield properties. The NPN uptake factors and PI fluorescence intensity increased. The problem of antimicrobial resistance has encouraged a re-evaluation of old antibiotics, with a view to repurposing them to combat newly emerging threats. HBPL-6 showed obvious bacteriostatic effects combined with erythromycin and rifampicin, exhibiting the potential to solve the problem of using old drugs by administering them using new methods. Moreover, the combination of 1.56 μg/mL of HBPL-6 and 2 μg/mL of colistin was fully synergistic for the inhibition of *S*. *typhimurium*; therefore, HBPL-6 could be used as a low-toxic, highly soluble, and high-yield polycation to reduce the emergence of colistin resistance in G^−^ bacteria. Though this study is not the first to demonstrate synergy between antimicrobial polymers and antibiotics, it contributes the exploration of hyperbranched polylysines used as cationic antimicrobial polymers for a wide range of G^−^ bacteria.

## Figures and Tables

**Figure 1 antibiotics-13-00217-f001:**
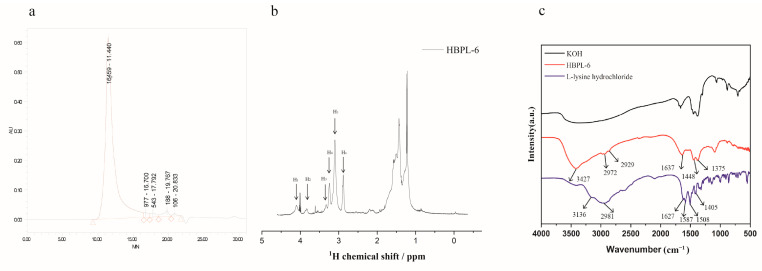
Structural characterization of polylysine. (**a**) The GPC spectra of HBPL-6 and the polymer show unimodal distributions. (**b**) ^1^H NMR of HBPL-6 in D_2_O. (**c**) FTIR spectra of L-lysine, KOH, and HBPL-6 in the 4000–500 cm^−1^ range.

**Figure 2 antibiotics-13-00217-f002:**
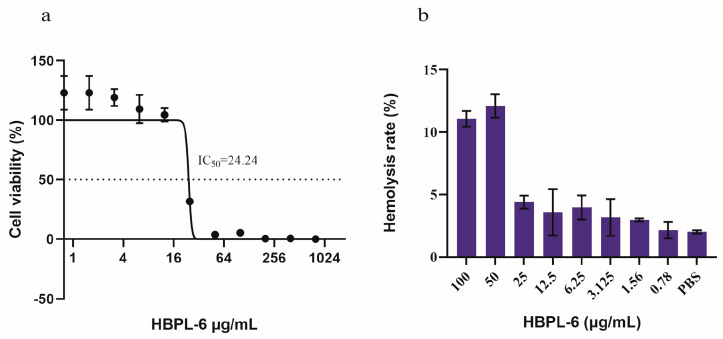
Cytotoxicity and hemolytic toxicity of HBPL-6. (**a**) Survival rate of monkey kidney Vero cells with different concentrations of HBPL-6 ranging from 0–100 μg/mL; IC_50_ = 24.24 μg/mL. (**b**) Hemolysis rate of sheep blood cells with the addition of the same concentration of HBPL-6 ranging from 0–100 μg/mL.

**Figure 3 antibiotics-13-00217-f003:**
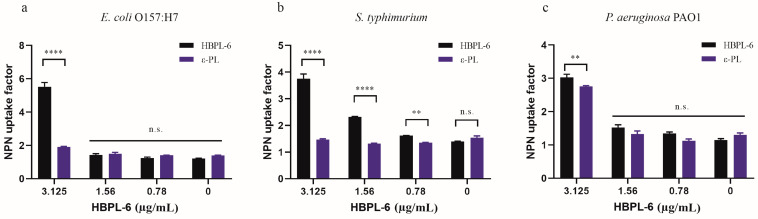
NPN fluorescence values of HBPL-6 compared with ε-PL. (**a**–**c**, respectively) *E. coli* O157:H7, *S. typhimurium*, and *P. aeruginosa* PAO1 treated with the addition of different concentrations (0.78–3.125 µg/mL) of HBPL-6 and ε-PL. (n.s.: not significant; *: *p* < 0.05, **: *p* < 0.01, ***: *p* < 0.001, ****: *p* < 0.0001).

**Figure 4 antibiotics-13-00217-f004:**
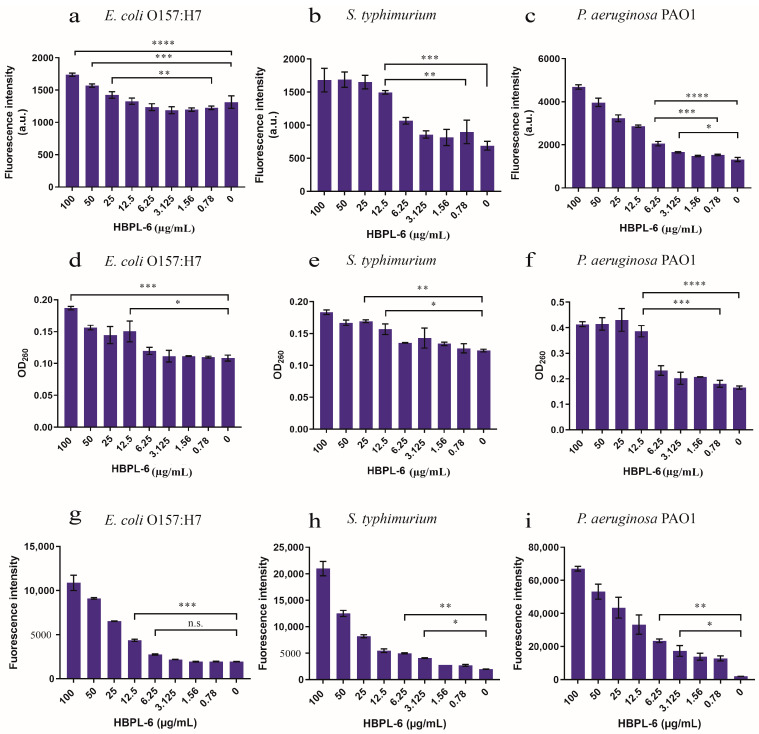
The assessment of outer-membrane permeabilization by the HBPL-6 of G^−^ bacteria. (**a**–**c**) PI fluorescence intensities of the additions of different concentrations of HBPL-6 (0.78–100 µg/mL) for the 30 min incubation periods of *E. coli* O157:H7, *S. typhimurium*, and *P. aeruginosa* PAO1, respectively. (**d**–**f**, respectively) Presence of 260 nm absorbing materials in the supernatants of *E. coli* O157:H7, *S. typhimurium*, and *P. aeruginosa* PAO1 treated with HBPL-6 (0.78–100 µg/mL). (**g**–**i**, respectively) Total ROS accumulation values for *E. coli* O157:H7, *S. typhimurium*, and *P. aeruginosa* PAO1 treated with HBPL-6 (0.78–100 µg/mL). The data are the average triplicates using a non-parametric one-way ANOVA test. (n.s.: not significant; *: *p* < 0.05, **: *p* < 0.01, ***: *p* < 0.001, ****: *p* < 0.0001).

**Figure 5 antibiotics-13-00217-f005:**
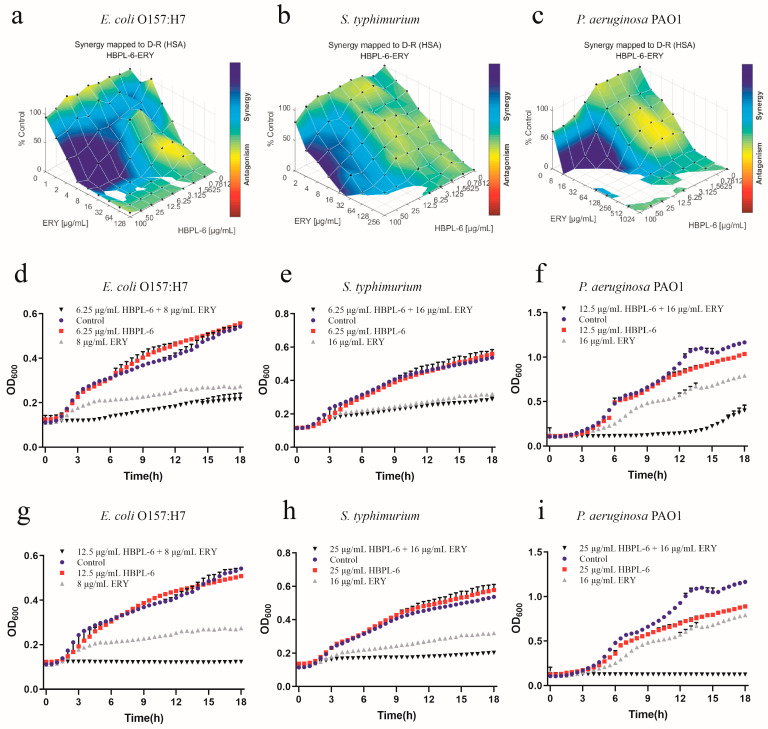
Plots were generated using the Combenefit program for the synergistic potentiation result for the use of HBPL-6 on erythromycin. (**a**) *E. coli* O157:H7 with HBPL-6 (0–100 μg/mL) and erythromycin (0–128 μg/mL). (**b**) *S. typhimurium* with HBPL-6 (0–100 μg/mL) and erythromycin (0–256 μg/mL). (**c**) *P. aeruginosa* PAO1 with HBPL-6 (0–100 μg/mL) and erythromycin (0–1024 μg/mL). Growth curves for single- and dual-drug combinations (**d**,**g**, respectively): *E. coli* O157:H7 with 6.25 and 12.5 μg/mL dose administrations of HBPL-6 and 8 μg/mL of erythromycin. (**e**,**h**) *S. typhimurium* with 6.25 and 25 μg/mL dose administrations of HBPL-6 and 16 μg/mL of erythromycin. (**f**,**i**) *P. aeruginosa* PAO1 with 12.5 and 25 μg/mL dose administrations of HBPL-6 and 16 μg/mL of erythromycin.

**Figure 6 antibiotics-13-00217-f006:**
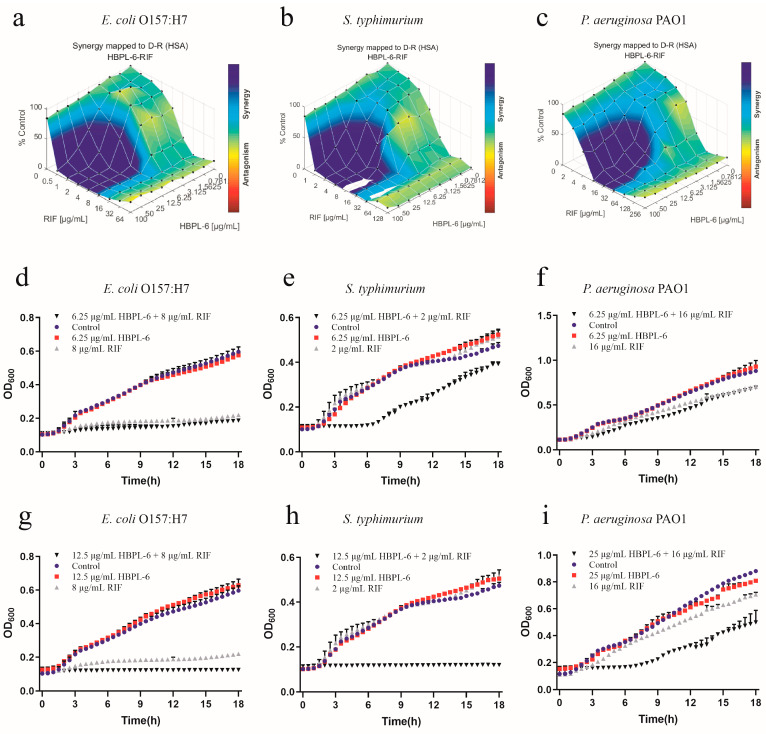
Plots were generated using the Combenefit program for the synergistic potentiation result of the use of HBPL-6 on rifampin. (**a**) *E. coli* O157:H7 with HBPL-6 (0–100 μg/mL) and rifampin (0–128 μg/mL). (**b**) *S. typhimurium* with HBPL-6 (0–100 μg/mL) and rifampin (0–64 μg/mL). (**c**) *P. aeruginosa* PAO1 with HBPL-6 (0–100 μg/mL) and rifampin (0–256 μg/mL). Growth curves for single- and dual-drug combinations (**d**,**g**, respectively): *E. coli* O157:H7 with 6.25 and 12.5 μg/mL doses of HBPL-6 and 8 μg/mL of rifampin. (**e**,**h**) *S. typhimurium* with 6.25 and 12.5 μg/mL doses of HBPL-6 and 2 μg/mL of rifampin. (**f**,**i**) *P. aeruginosa* PAO1 with 6.25 and 25 μg/mL doses of HBPL-6 and 16 μg/mL of rifampin.

**Figure 7 antibiotics-13-00217-f007:**
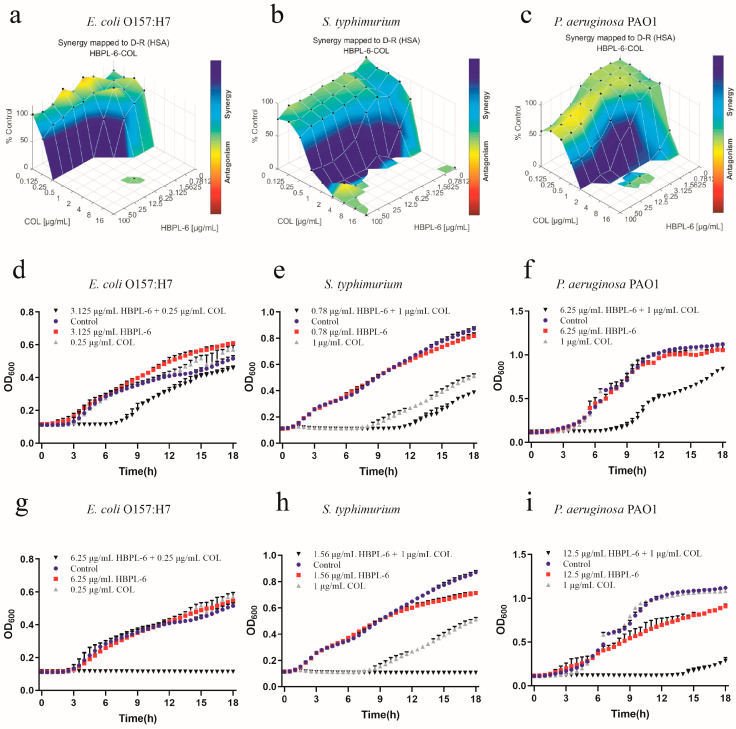
Plots were generated using the Combenefit program for the synergistic potentiation result for the use of HBPL-6 on colistin. (**a**) *E. coli* O157:H7 with HBPL-6 (0–100 μg/mL) and colistin (0–16 μg/mL). (**b**) *S. typhimurium* with HBPL-6 (0–100 μg/mL) and colistin (0–16 μg/mL). (**c**) *P. aeruginosa* PAO1 with HBPL-6 (0–100 μg/mL) and colistin (0–16 μg/mL). Growth curves for single- and dual-drug combinations (**d**,**g**, respectively): *E. coli* O157:H7 with 3.125 and 6.25 μg/mL doses of HBPL-6 and 0.25 μg/mL of colistin. (**e**,**h**) *S. typhimurium* with 0.78 and 1.56 μg/mL doses of HBPL-6 and 2 μg/mL of colistin. (**f**,**i**) *P. aeruginosa* PAO1 with 6.25 and 12.5 μg/mL doses of HBPL-6 and 1 μg/mL of colistin.

**Figure 8 antibiotics-13-00217-f008:**
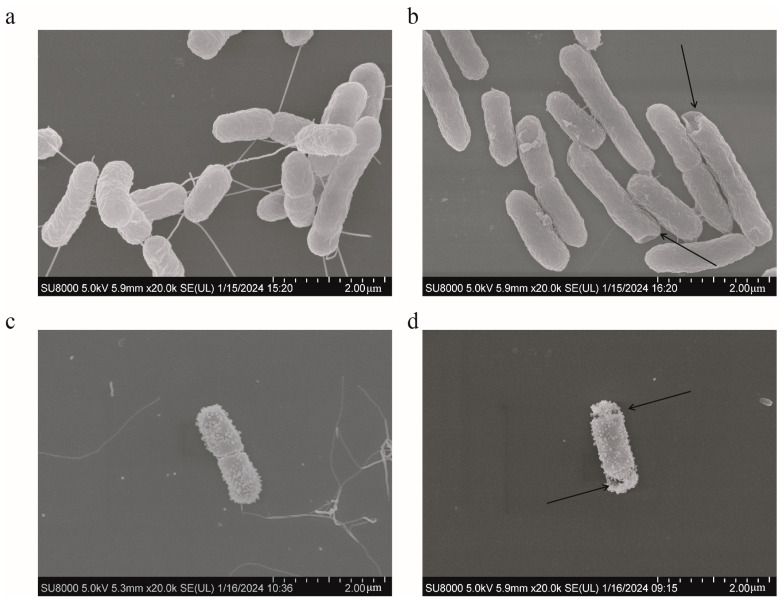
SEM of *S. typhimurium.* (**a**) Untreated; (**b**) treated with HBPL-6 at a dose of 12.5 μg/mL; (**c**) treated with colistin at a designated FIC value (1 μg/mL); (**d**) treated with HBPL-6 and colistin at the synergistic concentration.

**Table 1 antibiotics-13-00217-t001:** Fluorescence values obtained for the NPN uptake assay of *E. coli* O157:H7 following the addition of HBPL-6 in the range of 0.78–100 µg/mL.

Sample	NPN	Fluorescence Value	Subtracted Background Value	NPN Uptake Factor
Buffer	−	811.1666667		
Buffer	+	9696.666667	8885.50	1.00
Cell	−	965.5		
Cell	+	11,842.33333	10,876.83	1.22
Cell+HBPL-6 at 100 µg/mL	−	11,786.5		
Cell+HBPL-6 at 100 µg/mL	+	117,166.1667	105,379.67	11.86
Cell+HBPL-6 at 50 µg/mL	−	6989.833333		
Cell+HBPL-6 at 50 µg/mL	+	112,081.8333	105,092.00	11.83
Cell+HBPL-6 at 25 µg/mL	−	4238		
Cell+HBPL-6 at 25 µg/mL	+	85,813.83333	81,575.83	9.18
Cell+HBPL-6 at 12.5 µg/mL	−	2535.5		
Cell+HBPL-6 at 12.5 µg/mL	+	58,615.33333	56,079.83	6.31
Cell+HBPL-6 at 6.25 µg/mL	−	1969.166667		
Cell+HBPL-6 at 6.25 µg/mL	+	50,474.66667	48,505.50	5.46
Cell+HBPL-6 at 3.125 µg/mL	−	1350		
Cell+HBPL-6 at 3.125 µg/mL	+	47,191.83333	45,841.83333	5.16
Cell+HBPL-6 at 1.56 µg/mL	−	1216		
Cell+HBPL-6 at 1.56 µg/mL	+	14,067.16667	12,851.16667	1.45
Cell+HBPL-6 at 0.78 µg/mL	−	1063.666667		
Cell+HBPL-6 at 0.78 µg/mL	+	12,175.66667	11,112	1.25

In the table, + means 50 μL of 40 μmol/L N-phenyl-1-naphthylamine (NPN) (dissolved in HEPES buffer) was added to wells, and – means 50 μL of HEPES buffer without NPN.

**Table 2 antibiotics-13-00217-t002:** Fluorescence values obtained for an NPN uptake assay of *S. typhimurium* after the addition of HBPL-6 in the range of 0.78–100 µg/mL.

Sample	NPN	Fluorescence Value	Subtracted Background Value	NPN Uptake Factor
Buffer	−	779.5		
Buffer	+	5927.666667	5148.17	1.00
Cell	−	1604.833333		
Cell	+	8865.666667	7260.83	1.41
Cell+HBPL-6 at 100 µg/mL	−	12,767.66667		
Cell+HBPL-6 at 100 µg/mL	+	49,625	36,857.33	7.16
Cell+HBPL-6 at 50 µg/mL	−	7674		
Cell+HBPL-6 at 50 µg/mL	+	34,989.66667	27,315.67	5.31
Cell+HBPL-6 at 25 µg/mL	−	4776.666667		
Cell+HBPL-6 at 25 µg/mL	+	29,421.83333	24,645.17	4.79
Cell+HBPL-6 at 12.5 µg/mL	−	3278.5		
Cell+HBPL-6 at 12.5 µg/mL	+	25,181.33333	21,902.83	4.25
Cell+HBPL-6 at 6.25 µg/mL	−	2429.166667		
Cell+HBPL-6 at 6.25 µg/mL	+	23,741	21,311.83	4.14
Cell+HBPL-6 at 3.125 µg/mL	−	1967.833333		
Cell+HBPL-6 at 3.125 µg/mL	+	20,948.66667	18,980.83	3.69
Cell+HBPL-6 at 1.56 µg/mL	−	1740.166667		
Cell+HBPL-6 at 1.56 µg/mL	+	13,686.16667	11,946	2.32
Cell+HBPL-6 at 0.78 µg/mL	−	1701.833333		
Cell+HBPL-6 at 0.78 µg/mL	+	9985.666667	8283.83	1.61

**Table 3 antibiotics-13-00217-t003:** Fluorescence values obtained for an NPN uptake assay of *P. aeruginosa* PAO1 following the addition of HBPL-6 in the range of 0.78–100 µg/mL.

Sample	NPN	Fluorescence Value	Subtracted Background Value	NPN Uptake Factor
Buffer	−	816.1666667		
Buffer	+	9605.666667	8789.50	1.00
Cell	−	948		
Cell	+	11,021.33333	10,073.33	1.15
Cell+HBPL-6 at 100 µg/mL	−	13,756.16667		
Cell+HBPL-6 at 100 µg/mL	+	107,066.8333	93,310.67	10.62
Cell+HBPL-6 at 50 µg/mL	−	8544.833333		
Cell+HBPL-6 at 50 µg/mL	+	102,247.8333	93,703.00	10.66
Cell+HBPL-6 at 25 µg/mL	−	5086		
Cell+HBPL-6 at 25 µg/mL	+	90,774.83333	85,688.83	9.75
Cell+HBPL-6 at 12.5 µg/mL	−	3019.5		
Cell+HBPL-6 at 12.5 µg/mL	+	80,323.66667	77,304.17	8.80
Cell+HBPL-6 at 6.25 µg/mL	−	1934.833333		
Cell+HBPL-6 at 6.25 µg/mL	+	77,972.33333	53,770.83	6.12
Cell+HBPL-6 at 3.125 µg/mL	−	1418		
Cell+HBPL-6 at 3.125 µg/mL	+	25,619.5	24,201.5	2.75
Cell+HBPL-6 at 1.56 µg/mL	−	1172		
Cell+HBPL-6 at 1.56 µg/mL	+	14,529.83333	13,357.83333	1.52
Cell+HBPL-6 at 0.78 µg/mL	−	1046.666667		
Cell+HBPL-6 at 0.78 µg/mL	+	12,893.83333	11,847.16667	1.35

**Table 4 antibiotics-13-00217-t004:** Minimum inhibitory concentrations of HBPL-6 for representative Gram-negative bacteria.

Microorganism	MICs of HBPL-6 (μg/mL)
Gram-negative bacteria	
*E. coli* O157:H7	>2500
*S. typhimurium*	>2500
*P. aeruginosa* PAO1	>2500

**Table 5 antibiotics-13-00217-t005:** Minimum inhibitory concentrations of different antibiotics in the presence of HBPL-6 used to combat *S. typhimurium*.

Antibiotics	MICs of Antibiotics (μg/mL)	MICs of Antibiotics Supplemented with HBPL-6 (μg/mL)	FICIs * of Antibiotics
Rifampicin	32	1	0.03125
Erythromycin	128	16	0.125
Colistin	2	1	0.5
Minocycline	4	2	0.5
Tetracycline	32	16	0.5
Gentamicin	1	1	1
Neomycin	3.75	3.75	1
Tobramycin	3.75	3.75	1
Amikacin	1.5	1.5	1
Ampicillin	2	2	1

* FIC a = MICa in combination/MICa alone; FIC b = MICb in combination/MICb alone and FICI = FICa + FICb. a = HBPL-6; b = antibiotic. The FICIs are interpreted as follows: (1) a synergistic effect when the FICI ≤ 0.5; (2) an additive or indifferent effect when the FICI > 0.5 and ≤ 1; and (3) an antagonistic effect when the FICI > 1.

## Data Availability

All datasets generated for this study can be provided upon request.

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
