# Peer review of "Hyperbranched Polylysine Exhibits a Collaborative Enhancement of the Antibiotic Capacity to Kill Gram-Negative Pathogens"

_antibiotics, 2024, doi:10.3390/antibiotics13030217_

Round 1

Reviewer 1 Report

Comments and Suggestions for Authors

The study highlights the potential of HBPL-6 in resensitizing resistant pathogens and preventing the emergence of drug resistance, offering a novel strategy in the fight against bacterial infections.

Major objection:

MIC evaluation was performed only using standard strains, therefore results of the further evaluation can be questioned due to the lack of results linked with bacteria clinical isolates. A panel of Gram-negative strains is limited to the 3 strains. Additional tests should be performed using clinical isolates to support the thesis that HBPL-6 can enhance the effectiveness of certain antibiotics against Gram-negative bacteria.

Minor objections detected:

1. Cytotoxicity and hemolytic toxicity of HBPL-6 - "In conclusion, the cytotox- 121 icity and hemolytic of synthetic polycations generally increases with concentration." Please rephrase since the results presented do not support this conclusion.

2. In Table 2, FICI evaluation should be briefly explained (maybe below the table with an asterisk. Furthermore, besides methodology (placed in Methods) please add some comments on the results of FICI. Usually, this evaluation is intended to detect synergism in action. It should be explained which antibiotics were synergistic, additive, or showed indifference or antagonism. Please elaborate briefly in this section or section 2.5 should be connected directly with 2.3.

Author Response

Response letter

Ref; Manuscript ID: antibiotics-2808972

We would like to thank Editor and reviewers for their comments and suggestions on our manuscript. The revised manuscript includes continuous line numbers and the comments raised by the reviewer’s response point by point as shown in Red color in the revised version. The English revisions have been done in editing service.

Referee#1: 

Thank you for your time, we appreciate your comments and suggestions it will help to revise our manuscript. The raised comments were answered point by point in the revised manuscript accordingly.

Comments:

Major objection:

MIC evaluation was performed only using standard strains, therefore results of the further evaluation can be questioned due to the lack of results linked with bacteria clinical isolates. A panel of Gram-negative strains is limited to the 3 strains. Additional tests should be performed using clinical isolates to support the thesis that HBPL-6 can enhance the effectiveness of certain antibiotics against Gram-negative bacteria.

Response to major objection;

Thank you for comment. During the last few years, the increased incidence of gram-negative bacterial strains with decreased susceptibility, or resistance, to antibiotics appeared as a serious concern worldwide, with Salmonella Typhimurium, Escherichia coli, and Pseudomonas aeruginosa being the prominent causal agents. In this paper, these strains are used as representative organisms to provide a theoretical guide for the subsequent discovery of drug-resistant bacteria in clinical practice. This study needs to be followed up by clinical isolates to further validate the consistency of synergistic efficacy. The combination has been shown to be effective in Salmonella pullorum, CVCC 1789 (purchased from National Center for Veterinary Culture Collection, China) and will be published in a subsequent article. Besides, fully susceptible strains can be used to investigate whether the conjugation of antibiotics and adjuvant reduces the emergence of the least sensitive first-step resistance mutants in bacterial populations.

References;

  1. Nahar, S.; Ha, A.J.-w.; Byun, K.-H.; Hossain, M.I.; Mizan, M.F.R.; Ha, S.-D. Efficacy of flavourzyme against Salmonella Typhimurium, Escherichia coli, and Pseudomonas aeruginosa biofilms on food-contact surfaces. International Journal of Food Microbiology 2021, 336, 108897, doi:https://doi.org/10.1016/j.ijfoodmicro.2020.108897.
  2. Pasquali, F.; Manfreda, G. Mutant prevention concentration of ciprofloxacin and enrofloxacin against Escherichia coli, Salmonella Typhimurium and Pseudomonas aeruginosa. Veterinary Microbiology 2007, 119, 304-310, doi:https://doi.org/10.1016/j.vetmic.2006.08.018.

Minor objections detected:

  1. Cytotoxicity and hemolytic toxicity of HBPL-6 - "In conclusion, the cytotoxicity and hemolytic of synthetic polycations generally increases with concentration." Please rephrase since the results presented do not support this conclusion.

Response to comment 1;

Thank you for the suggestion. The sentence “In conclusion, the cytotoxicity and hemolytic of synthetic polycations generally increases with concentration.” has been changed to “Based on these results, it can be observed that the cytotoxicity and hemolytic properties of HBPL-6 generally increase with the concentration.” See lines 133-134.

  1. In Table 2, FICI evaluation should be briefly explained (maybe below the table with an asterisk. Furthermore, besides methodology (placed in Methods) please add some comments on the results of FICI. Usually, this evaluation is intended to detect synergism in action. It should be explained which antibiotics were synergistic, additive, or showed indifference or antagonism. Please elaborate briefly in this section or section 2.5 should be connected directly with 2.3.

Response to comment 2;

Thank you for comment. (1) The FICI evaluation has been briefly explained below the table with an asterisk (lines 228-230) and the detailed comments of the results for FICI has been added to 2.4.2 (lines 212-219). And based on the results available, it was analysed in the discussion part of the reasons for the different effects among (lines 386-389 and lines 414-422). (2) The section 2.3 has been modified to connect directly to part 2.5. See 2.4 MIC Determination and Checkerboard Synergy Test.

References;

  1. Gorityala, B.K.; Guchhait, G.; Fernando, D.M.; Deo, S.; McKenna, S.A.; Zhanel, G.G.; Kumar, A.; Schweizer, F. Adjuvants Based on Hybrid Antibiotics Overcome Resistance in Pseudomonas aeruginosa and Enhance Fluoroquinolone Efficacy. Angewandte Chemie International Edition 2016, 55, 555-559, doi:https://doi.org/10.1002/anie.201508330.
  2. Lyu, Y.; Yang, X.; Goswami, S.; Gorityala, B.K.; Idowu, T.; Domalaon, R.; Zhanel, G.G.; Shan, A.; Schweizer, F. Amphiphilic Tobramycin–Lysine Conjugates Sensitize Multidrug Resistant Gram-Negative Bacteria to Rifampicin and Minocycline. Journal of Medicinal Chemistry 2017, 60, 3684-3702, doi:10.1021/acs.jmedchem.6b01742.
  3. Schweizer, F. Enhancing uptake of antibiotics into Gram-negative bacteria using nonribosome-targeting aminoglycoside-based adjuvants. Future medicinal chemistry 2019, 11, 1519-1522, doi:10.4155/fmc-2019-0131.

Reviewer 2 Report

Comments and Suggestions for Authors

Drug resistance has been an important global public health concern and many efforts have been on seeking and developing new strategies against the antibiotic resistance threat to combat bacterial infections. In this study, Gong, Y. et al. reported a cationic hyperbranched polylysine (HBPL-6) that was synthesized by one pot polymerization method with properties including low-toxic, high-solubility and high-yield and investigated its antibacterial activities, especially its synergistic antibacterial activity against gram-negative bacteria. This study provides an example of using hyperbranched polylysine as cationic antimicrobial polymers against a wide range of gram-negative bacteria. The followings are some concerns and suggestions for the authors’ consideration:        

1.    Table 2, several antibiotics, like Gentamicin, Neomycin, Tobramycin, etc. target bacterial ribosomes/protein synthesis which are the intracellular targets, however, there is no synergistical effect when combining with BHPL-6. What could be the possible reasons? It would be better to have some discussions in the text.

2.    Regarding the mode of action of HBPL-6, it suggests providing direct data to showing HBPL-6 disrupts membranes and investigate the peptides effect on bacterial cellular morphology, for example, SEM or TEM imaging to observe ultrastructural changes of bacterial membrane integrity.

3.    Line 74, “alanineto” should be “alanine to”.Page 11, Line 258-261 (the first paragraph in Discussion Section), these are the review comments, not the author’s discussion, therefore, this paragraph should be deleted.

4.   Line 86, the subsection title is “2.1. Synthesis and characterization of HBPL-6”, however, there is no description of the synthesis. Suggest changing to “2.1. Characterization of synthetic HBPL-6”. 

Comments on the Quality of English Language

Overall, it looks good, maybe minor editing is required

Author Response

Response letter

Ref; Manuscript ID: antibiotics-2808972

We would like to thank Editor and reviewers for their comments and suggestions on our manuscript. The revised manuscript includes continuous line numbers and the comments raised by the reviewer’s response point by point as shown in Red color in the revised version. The English revisions have been done in editing service.

Referee#2: 

Thank you for your time, we appreciate your comments and suggestions it will help to revise our manuscript. The raised comments were answered point by point in the revised manuscript accordingly.

Comments:

Drug resistance has been an important global public health concern and many efforts have been on seeking and developing new strategies against the antibiotic resistance threat to combat bacterial infections. In this study, Gong, Y. et al. reported a cationic hyperbranched polylysine (HBPL-6) that was synthesized by one pot polymerization method with properties including low-toxic, high-solubility and high-yield and investigated its antibacterial activities, especially its synergistic antibacterial activity against gram-negative bacteria. This study provides an example of using hyperbranched polylysine as cationic antimicrobial polymers against a wide range of gram-negative bacteria. The followings are some concerns and suggestions for the authors’ consideration:

  1. Table 2, several antibiotics, like Gentamicin, Neomycin, Tobramycin, etc. target bacterial ribosomes/protein synthesis which are the intracellular targets, however, there is no synergistical effect when combining with BHPL-6. What could be the possible reasons? It would be better to have some discussions in the text.

Response to comment 1;

Thank you for comment. As you suggested, we have added some discussions of the possible reasons. “However, it also shows the moderate effects of aminoglycoside antibiotics, e.g., gentamicin, neomycin, tobramycin, and amikacin, when combined with BHPL-6 (FICI = 1). Recently conducted research demonstrates that appending hydrophobic moieties onto aminoglycosides, like tobramycin, can generate amphiphilic aminoglycosides with weak or no protein translation inhibitory effects, but can also exhibit potent OM-disrupting and IM-uncoupling properties against P. aeruginosa [1,2]. Based on the results of this study, we speculated that the possible cause of the indifferent effect was that the outer-membrane disruption of tobramycin could be uncoupled from its ribosomal effects [3].” See lines 414-422.

References;

  1. Gorityala, B.K.; Guchhait, G.; Fernando, D.M.; Deo, S.; McKenna, S.A.; Zhanel, G.G.; Kumar, A.; Schweizer, F. Adjuvants Based on Hybrid Antibiotics Overcome Resistance in Pseudomonas aeruginosa and Enhance Fluoroquinolone Efficacy. Angewandte Chemie International Edition 2016, 55, 555-559, doi:https://doi.org/10.1002/anie.201508330.
  2. Lyu, Y.; Yang, X.; Goswami, S.; Gorityala, B.K.; Idowu, T.; Domalaon, R.; Zhanel, G.G.; Shan, A.; Schweizer, F. Amphiphilic Tobramycin–Lysine Conjugates Sensitize Multidrug Resistant Gram-Negative Bacteria to Rifampicin and Minocycline. Journal of Medicinal Chemistry 2017, 60, 3684-3702, doi:10.1021/acs.jmedchem.6b01742.
  3. Schweizer, F. Enhancing uptake of antibiotics into Gram-negative bacteria using nonribosome-targeting aminoglycoside-based adjuvants. Future medicinal chemistry 2019, 11, 1519-1522, doi:10.4155/fmc-2019-0131.

  1. Regarding the mode of action of HBPL-6, it suggests providing direct data to showing HBPL-6 disrupts membranes and investigate the peptides effect on bacterial cellular morphology, for example, SEM or TEM imaging to observe ultrastructural changes of bacterial membrane integrity.

Response to comment 2;

As you suggested, we have added the SEM results. It was used to observe the disruption of membranes during the treatment of HBPL-6 and colistin for S. typhimurium. What’s more, the corresponding descriptions of the results were added “The bacteria in the control group had rod-shaped cells with relatively smooth surfaces and intact cell membranes (Fig. 8 a). No bacterial changes were observed in the FIC values of colistin (Fig. 8 c)., however surface wrinkling occurred following the addition of 12.5 μg/mL of HBPL-6 (Fig. 8 b). For the combined treatment, the alterations occurred on the bacterial surface and the cell membrane ruptured with pores (Fig. 8 d)”. See Figure 8 and Line 307-312.

  1. Line 74, “alanineto” should be “alanine to”.Page 11, Line 258-261 (the first paragraph in Discussion Section), these are the review comments, not the author’s discussion,therefore, this paragraph should be deleted.

Response to comment 3;

Thank you for the suggestion. The phrase “alanineto” has been deleted in article revisions. Furthermore, Line 258-261 (the first paragraph in Discussion Section), they appeared to be the original note from a manuscript template. So we have deleted the paragraph.

  1. Line 86, the subsection title is “2.1. Synthesis and characterization of HBPL-6”, however, there is no description of the synthesis. Suggest changing to “2.1. Characterization of synthetic HBPL-6”.

Response to comment 4;

Thank you for the suggestion. The title “2.1. Synthesis and characterization of HBPL-6” has been changed to “2.1. Characterization of synthetic HBPL-6”. See lines 99.

Reviewer 3 Report

Comments and Suggestions for Authors

See file attached. 

Comments on the Quality of English Language

Must be improved.

Author Response

Response letter

Ref; Manuscript ID: antibiotics-2808972

We would like to thank Editor and reviewers for their comments and suggestions on our manuscript. The revised manuscript includes continuous line numbers and the comments raised by the reviewer’s response point by point as shown in Red color in the revised version. The English revisions have been done in editing service.

Referee#3: 

Thank you for your time, we appreciate your comments and suggestions it will help to revise our manuscript. The raised comments were answered point by point in the revised manuscript accordingly.

Comments:

  1. The literature review in the INTRODUCTION section does not flow, likely related to the English language. To give an example, line 38 says “Colistin is always used to treat gastrointestinal diseases of animals.” Why would an antibiotic be used to treat gastrointestinal inflammation?? There are a few other sentences in this section appeared to be incomplete and misleading readers. Another example in lines 41-42, where it mentioned about the ‘MCR gene’, but it then jumped to a completely different topic about the Colistin being the last resort in human medicine.

Response to comment 1;

As you suggested, the sentence “Colistin is always used to treat gastrointestinal diseases of animals.” has been changed to “Intestinal infections caused as a result of Gram-negative (G-) bacteria are the main reasons for the use of colistin in livestock [1]. Furthermore, it has been reported in the literature that colistin is the most-often prescribed antibiotic used to treat diarrhea caused by E. coli, at 40% for pigs and 30% for cattle [2].” to illustrate why colistin be used to treat intestinal infections. See lines 41-44.The not flow sentences has been changed “The use of colistin in animals has been limited by the emergence of resistant bacteria as the colistin dosage has increased [3]. ” See lines 44-45.

References;

  1. Rhouma, M.; Fairbrother, J.M.; Beaudry, F.; Letellier, A. Post weaning diarrhea in pigs: risk factors and non-colistin-based control strategies. Acta veterinaria Scandinavica 2017, 59, 31, doi:10.1186/s13028-017-0299-7.
  2. Jansen, W.; van Hout, J.; Wiegel, J.; Iatridou, D.; Chantziaras, I.; De Briyne, N. Colistin Use in European Livestock: Veterinary Field Data on Trends and Perspectives for Further Reduction. Veterinary Sciences 2022, 9, 650.
  3. Catry, B.; Cavaleri, M.; Baptiste, K.; Grave, K.; Grein, K.; Holm, A.; Jukes, H.; Liebana, E.; Navas, A.L.; Mackay, D.; et al. Use of colistin-containing products within the European Union and European Economic Area (EU/EEA): development of resistance in animals and possible impact on human and animal health. International Journal of Antimicrobial Agents 2015, 46, 297-306, doi:https://doi.org/10.1016/j.ijantimicag.2015.06.005.
  4. I would suggest adding representative structures for some of the polymer in the INTRODUCTION section that are related to this study, i.e. the (i) basic Polylysine (ii) Dendritic version (iii) poly-L-Lysine (pLK) and the (iv) actual HBPL-6. This would help to elaborate the results & discussion parts where the synthetic and analytical chemistry were reported.

Response to comment 2;

As you suggested, we have added the sentence “Poly-L-lysine mainly consists of ε-Poly-L-lysine (ε-PL) and α-Poly-L-lysine (PLL), which are linear chain polymers. Dendritic poly-L-lysine (DPL) is a synthetic nanomaterial, and it has perfect monodispersed macromolecules with a regular, defect-free branched architecture [4]. Hyperbranched polylysine (HBPL) is structurally related to dendritic polylysine, but with a random, branched structure and both ε-PL and PLL units [5]. ” in the introduction. See lines 68-73.

References;

  1. Scholl, M.; Nguyen, T.Q.; Bruchmann, B.; Klok, H.-A. Controlling Polymer Architecture in the Thermal Hyperbranched Polymerization of l-Lysine. Macromolecules 2007, 40, 5726-5734, doi:10.1021/ma070494l.
  2. Kadlecova, Z.; Rajendra, Y.; Matasci, M.; Baldi, L.; Hacker, D.L.; Wurm, F.M.; Klok, H.-A. DNA delivery with hyperbranched polylysine: A comparative study with linear and dendritic polylysine. Journal of Controlled Release 2013, 169, 276-288, doi:https://doi.org/10.1016/j.jconrel.2013.01.019.
  3. Raw spectra as currently shown in Figure 1a-1f can be kept as supplementary information if authors wish to, as these raw figures are not displaying useful information and reflecting to the details described in the results, discussion and methodology sections. Particularly in …
  • Figure 1a – the numerical values currently labelled in the gel chromatography spectrum shows nothing related to the Mn (3359 & 13159 g/mol), which is described in section 2.1.
  • Figure 1b – technically, a C-H correlation 2D NMR spectrum (HSQC) should be provided to indicate the proton environment of the hyperbranched property. The authors currently using this spectrum to indicate the presence of the dendritic units, N-linked and terminal units, which is rather peculiar. If this is truly the case, authors will need to further process this spectrum, of which be displaying molecular structure indicating the corresponding signals and labelled within the NMR spectrum, a reference would also be required to support this interpretation. Also, in the methodology, authors clamed that the DB and ANB were determined using NMR technique, such analysis is not shown in the paper. Please provide proton integral information where the authors had claimed to have used for to calculate the DB and ANB.
  • Figure 1c – again, the IR spectrum should be further processed to include the signals which are described in the main text.
  • Figure 1d – the main script should not contain raw spectrum of starting material, this figure and Figure 1e can simply indicated the starting material (mono-lysine HCl) is no longer a monomer. 2 • As per the note above, X-ray powder diffraction is used for amorphous solid, hence X-ray crystallography techniques would be used for studying crystalline samples. Hence, Figure 1d and 1e in this report cannot be used to identify whether the sample is crystalline or amorphous (Line 103-107). Again, if there is any evidence to support authors interpretation, please provide corresponding reference. If not, I would suggest removing the XRD results and simply note that a reaction of the Mono-Lysine has taken place.
  • Figure 1f is again a peculiar one, it is uncertain how these organic polymers would have optical properties, as their structures do not display any conjugation (C=C bonds). If there is evident supporting such Fluorescent properties for these polymers, please provide the corresponding reference. In general, the presentation of each figure in this manuscript should have been thought through and appropriately processed to reflect on the corresponding text described in the main script.

Response to comment 3;

Thank you for your suggestions. We have made additions and deletions to the figures and made changes in the results and discussion sections.

  • Figure 1a

The GPC traces showed that the elution peak of HBPL-6 is at 11.44 min and the MP is 16459 using Waters 2695 HPLC (Waters Corporation, Milford, MA, USA). Standards used for molecular weight calibration curves (purchased from Sigma): 1. Cytochrome C (Mw = 12384); 2. Peptidase (Mw = 6511); 3. Mycopeptide (Mw = 1422); 4. Ethionine-ethionine-tyrosine-arginine (Mw = 451); 5. Ethionine-Ethionine-Ethionine (Mw = 189). Subsequently, the Mw, Mn, Mp were obtained by slicing the spectra and using the statistical equation by GPC software of Empower workstations. There are two types of integrals as follows, and since it is a polydisperse hyperbranched polymer, we chose the second type of integration.

Time

Mn

Mw

MP

Integral area

% aera

1

11.440

9311

13882

16459

49938240

94.60

2

16.700

878

893

977

706696

1.34

3

17.792

508

526

543

763230

1.45

4

19.767

210

222

188

1073039

2.03

5

20.833

104

106

106

306835

0.58

移动时间

Mn

Mw

MP

面积

% 面积

1

11.440

3359

13159

16459

52776304

100.00

  • Figure 1b

The C-H correlation 2D NMR spectrum (HSQC) has been provided to indicate the proton environment of the hyperbranched property. See Figure S1.

See Line 465-469. The mole fractions of Nα-linked linear units (Lα), Nε-linked linear units (Lε), dendritic (D) units, and terminal (T) structural units were estimated by 1H NMR spectroscopy using the integrals of the α-CH signals corresponding to the different structural units. From the mole fractions of the structural units, the DB and the ANB were calculated following the definitions:

As shown in Figure 1 b, the HBPL-6 appears at 4.11 (H1), 3.88 (H2), 3.36 (H3), and 3.26 (H4), 3.10 (H5), 2.89 (H6) ppm corresponding to CαH protons in dendritic (D), Nα-linked linear (Lα), terminal (T), Nε-linked linear (Lε) units, CεH2 group next to an amide bond (dendritic and ε-linear structural unit) and CεH2 group in α-linear and terminal structural unit.

As shown below, our 1H NMR spectroscopy is similar to the previous research reported [6].

References;

  1. Alazzo, A.; Lovato, T.; Collins, H.; Taresco, V.; Stolnik, S.; Soliman, M.; Spriggs, K.; Alexander, C. Structural variations in hyperbranched polymers prepared via thermal polycondensation of lysine and histidine and their effects on DNA delivery. Journal of Interdisciplinary Nanomedicine 2018, 3, 38-54, doi:https://doi.org/10.1002/jin2.36.
  • Figure 1c

Thank you for comment. The signals have been further processed which are described in the main text. See Figure 1 c. Besides, we have modified the spectrum results in detail. See Line 108-119.

  • Figure 1d e f

Thank you for comment. We have deleted figure1 d e f.

  1. I would suggest authors to keep the MIC results together. As it currently stands, the MIC screening results in section 2.3, showing not only antimicrobial effect of the HBPL-6, but also the different antibiotic in the presence of the HBPL-6 (Table 2), which is well separated from the main MIC results in section 2.5. In relation to this part, the weight information for HBPL-6 which was used to supplement the various antibiotics was missing when these were tested against typhimurium(Table 2).

Response to comment 4;

Thank you for your comment. As you suggested, we have connected 2.3 section with 2.5 section as “2.4 MIC Determination and Checkerboard Synergy Test” in revised manuscript. So the MIC results used to supple the various antibiotics was connected with follow-up relevant checkerboard results. See Line 195-232.

  1. Table 3, 4 and 5, another presentation issue, it is not clear what the ‘-‘ and ‘+’ symbols for under the NPN column in these tables and is it necessary to show all the ‘subtract background values’, shouldn’t there be a better presentation for these results rather than listing all the raw data in the tables?

Response to comment 5;

Thank you for comment. As the reference shown, it is a traditional presentation. In the table, ‘-’ and ‘+’ symbols mean with addition of NPN and no addition of NPN, respectively. In the follow research, it used same presentation[7].

References;

  1. 7. Helander, I.M.; Mattila‐Sandholm, T. Fluorometric assessment of Gram‐negative bacterial permeabilization. Journal of Applied Microbiology 2000, 88, 213-219, doi:10.1046/j.1365-2672.2000.00971.x.
  2. Figure 3-7, please provide chart title (perhaps next to the chart label a, b & c etc) in each chart, so that it helps reader to easily visualise compare the results.

Response to comment 6;

Thank you for comment. We have provided chart title in each chart. See in Figure 3-7.

  1. Line 258-261, please remove these lines, they appeared to be the original note from a manuscript template.

Response to comment 7;

Thank you for comment. We have removed it.

  1. Line 272, the phrase ‘… less water-insoluble..’, does authors mean ‘more water soluble’??

Response to comment 8;

Thank you for comment. The “less water-insoluble” has been changed to “more water soluble”. See Line 333-334.

  1. The DISCUSSION section is the most confusing part of this manuscript, as it reads like a literature review (e.g. line 281-306) rather than a discussion of author’s results from this research work

Response to comment 9;

Thank you for comment. This part of the discussion was re-drawn based on the results, see Line 342-356.

Round 2

Reviewer 2 Report

Comments and Suggestions for Authors

All concerns has been addressed in the revision.